# THE MULTIPLE SUBNETWORK HYPOTHESIS
## ENABLING MULTIDOMAIN LEARNING BY ISOLATING TASK-SPECIFIC SUBNETWORKS IN FEEDFORWARD NEURAL NETWORKS

## ABSTRACT

Neural networks have seen an explosion of usage and research in the past decade, particularly within the domains of computer vision and natural language processing. However, only recently have advancements in neural networks yielded performance improvements beyond narrow applications and translated to expanded multitask models capable of generalizing across multiple data types and modalities. Simultaneously, it has been shown that neural networks are overparameterized to a high degree, and pruning techniques have proved capable of significantly reducing the number of active weights within the network while largely preserving performance. In this work, we identify a methodology and network representational structure which allows a pruned network to employ previously unused weights to learn subsequent tasks. We employ these methodologies on well-known benchmarking datasets for testing purposes and show that networks trained using our approaches are able to learn multiple tasks, which may be related or unrelated, in parallel or in sequence without sacrificing performance on any task or exhibiting catastrophic forgetting.

## 1 INTRODUCTION

It is well known and documented that artificial neural networks (ANNs) are often overparameterized, resulting in computational inefficiency (LeCun et al., 1990; Liu et al., 2019). Applying unstructured pruning to an ANN pares down the number of active weights by identifying which subset of weights in an ANN is most important for the model's predictive performance and discarding those which are less important or even entirely unnecessary. This technique has been shown to reduce the computational cost of using and storing a model without necessarily affecting model accuracy (Frankle & Carbin, 2019; Suzuki et al., 2001; Han et al., 2016; Lis et al., 2019; Wang et al., 2021). This phenomenon naturally raises a corollary question: are pruned weights entirely useless, or could weights identified as being unnecessary for one task be retained and used to learn other tasks? Further, can the ANN's performance on learned tasks be preserved while these weights learn to perform new tasks?

The exploration of these questions leads us to propose the *multitple subnetwork hypothesis* – that a dense, randomly-initialized feedforward ANN contains within its architecture multiple disjoint subnetworks which can be utilized together to learn and make accurate predictions on multiple tasks, regardless of the degree of similarity between tasks or input types. Instead of focusing on matching or surpassing state of the art results on the data sets and tasks presented in this study, we focus on testing the multiple subnetwork hypothesis on a set of standardized network architectures and compare multitask model performance with traditionally trained, single-task models of identical architectures.

An obstacle to developing multitask ANNs is the tendency of ANNs to exhibit catastrophic forgetting (CF), during which they destroy internal state representations used in learning previously acquired tasks when learning a new task (French, 1999; Goodfellow et al., 2013; Pfülb & Gepperth, 2019)). CF is especially pronounced in continuous learning paradigms with a low degree of intertask relatedness (Aljundi et al., 2017; Ma et al., 2018; Masana et al., 2021), posing a challenge to creating multitask models able to perform prediction or classification across very different tasks, input data types, or input shapes.

Isolating parameters for a given task, thereby fixing parameter subsets for a given task, is a means of minimizing the effects of CF in a continuous learning context (Delange et al., 2021). Parameter isolation methods localize the performance of learned tasks over a physically (Yoon et al., 2018; Rusu et al., 2016; Mallya & Lazebnik, 2018) or logically (Liu et al., 2019; Serra et al., 2018; Mallya et al., 2018) isolated region of an ANN, reducing or eliminating the interaction of weights responsible for a given task, therefore minimizing the opportunity for the acquisition of new tasks to interfere with the ability of an ANN to continue to perform previously learned tasks. While the isolation of subnetworks is the surest way to prevent interference arising from learning new tasks, this method can scale poorly under memory and computational constraints and therefore demands a flexible approach to limiting network capacity (Yoon et al., 2018) or minimizing the size of the network utilized for each task. Our methodology aims to negate the high computational cost of parameter isolation by representing multidomain multitask models as sparse tensors, thereby minimizing their computational footprint. In this work, we demonstrate the implications of the multiple subnetwork hypothesis through a set of experiments, the results of which demonstrate that our model training procedure and multitask representational structure can be used to create multiclass models with reduced computational footprints which are capable of overcoming CF.

## 2 METHODOLOGY

In this section, we will discuss how we enable a single model to learn multiple tasks across multiple domains and datatypes through both a task-specific weight representational structure and a modified training procedure which selects disjoint subsets of network weights for each individual task.

### 2.1 MULTITASK REPRESENTATIONAL STRUCTURE

The weights of fully connected neural network layers traditionally consist of a kernel tensor, $k$, of shape $m \times n$, and a bias vector, $b$, of length $n$, where $m$ denotes the number of columns in the input data and $n$ denotes the number of neurons within the layer. To process input data $\bar{x}$, the decision function is thus $\Phi(k\bar{x} + b)$, where $\Phi$ denotes the layer's activation function. Suppose then that there are inputs from two different distributions, $\bar{x}_1$ and $\bar{x}_2$. Given the two-dimensional structure of the kernel tensor in this traditional representation, there is no way to alter the decision function given the distribution currently selected.

To address this problem, we add a new dimension, $t$, to the kernel tensor and the bias vector within the layer to denote which distribution or task the input data belongs to, leaving the kernel tensor with a shape of $t \times m \times n$ and transforming the bias vector into a bias matrix of shape $t \times n$. The decision function for this new layer is thus altered to that in Equation 1.

$$F(\bar{x}, i) = \Phi(k_i\bar{x} + b_i) \tag{1}$$

We take a similar approach for convolutional layers. For a two-dimensional convolutions with color channels, the kernel is of shape $s_1 \times s_2 \times c \times f$, where $s_1$ and $s_2$ denote the height and width of the convolutional filters, $c$ represents the number of channels in the input, and $f$ corresponds to the number of filters. The bias vector in this scenario has length $f$. In our multitask representation of the convolutional layer, we once more add a new dimension to the front of both the kernel tensor and the bias vector to denote the task.

### 2.2 SPARSIFICATION AND SUBNETWORK IDENTIFICATION

While the multitask representational structure described above could theoretically be used to enable a network to learn multiple tasks as-is, it alone does not help us answer our hypothesis. Alone, this structure allows multiple networks to be combined into a single model, as the number of parameters within each layer is increased by a factor of the number of tasks. To fully test our multiple subnetwork hypothesis, we devised the Reduction of Sub-Network Neuroplasticity (RSN2) training procedure, which ensures that only one weight along the task dimension is active for all other fixed indices in multitask layers. In other words, due to the nature of RSN2's pruning schema, no two fixed weights along the task dimension, $t$, are active after pruning.

---

**Algorithm 1** RSN2 Training Procedure

---

**Input:** Neural Network $F(x)$, Training data for multiple tasks, $X_i, Y_i, i \in \{1, 2, .., N\}$
**for** $i \in \{1, 2, .., N\}$ **do**
    Choose $p_i, p_i \in (0, 100)$, ensuring $\sum_{i=1}^{N} p_i <= 100$
    Mask all weights which have been used for previous tasks
    Unmask all weights which have not been used for previous tasks
    Select $(x_i, y_i)$ as a subset of training data and labels for the current task
    Calculate weight gradients with respect to $(x_i, y_i)$
    Set all masked weight gradients to 0
    Identify the quantity $q_i$ such that $p\%$ of weight gradients with respect to $(x_i, y_i)$ is less than $q_i$
    Deactivate all weights with gradients less than $q_i$ and do not allow them to train
    Train the network as usual, only altering identified active weights and keeping all other weights masked
    Save all weight values for the specified task
**end for**
At inference time for task $i$, unmask only stored weights for task $i$ and perform inference as usual

---

$$I(x) = \left\{ \begin{array}{ll} 1, & \text{if } x \neq 0 \\ 0, & \text{otherwise} \end{array} \right\} \tag{2}$$

Mathematically, we utilize the indicator function from Equation 2 as an activation function to indicate whether an individual weight is active or inactive within a layer. Using this function, we can therefore demonstrate whether disjoint subnetworks are active for all tasks by taking the sum of the indicator function across the task dimension for all other fixed indices. If for all fixed indices, $b$ and $c$, the condition present in Equation 3 holds, then each weight is only active once across all tasks, meaning the task dimension adds no additional active weights to the network layer.

$$\sum_{a=1}^{t} I(k_{a,b,c}) \leq 1; \ b, c \text{ fixed} \tag{3}$$

We utilize binary masking in our training procedure to impose these conditions. These masks therefore ensure only select weights are active during training through the bitwise multiplication of the weight value and the mask value.

For weight selection and pruning, we utilize a gradient-based approach to identify which weights are to be selected for each task. By performing pruning in this manner, all pruning is done in a one-shot manner after initialization rather than iteratively as the model trains, with the pruning rate a hyperparameter selected at training time. The entirety of the RSN2 training procedure for a single network layer can be found in Algorithm 1. For performing inference, the masks used for training can be removed, thus reducing the number of active weights to a maximum value of $t*p*W$, where $t$ represents the number of tasks, $p$ the maximum proportion of weights active for any one task, and $W$ the number of weights which would be present in the network had a traditional, one-task structure been used. Note that to satisfy Equation 1, $t * p \leq 1$. As a result, the RSN2 procedure, when coupled with the multitask representational structure defined above, identifies and separates disjoint subnetworks within a traditional network to allow each to individually perform separate tasks. Due to the lack of interaction between these subnetworks in the resulting model, similarity between tasks is not required, and no transfer learning will occur between tasks. For a visual representation of the pruning methodology, we refer the reader to Appendix A.

## 2.3 EXPERIMENTS

We test our methodologies across a variety of experiments utilizing multiple datasets and various model architectures to test the multiple subnetwork hypothesis and RSN2 training procedure. In our first experiment, we test the merits of the hypothesis by identifying whether a single network can learn the same task multiple times. To test this, we create a single convolutional multitask network and isolate five separate subnetworks within it, each utilizing a disjoint 20% of the weights within

the overall network architecture. We then train each of these subnetworks on the Fashion MNIST dataset (Xiao et al., 2017). The performances of each of these subnetworks, as well as the ensemble of all five subnetworks, are then compared to a the performance of a dedicated network of identical architecture.

For our second experiment, we train a fully-connected network to perform both the MNIST Digit Recognition (LeCun & Cortes, 2010) and Fashion MNIST tasks and compare the performance of a single network to two dedicated networks. This experiment is the first we perform to test whether a single network can truly learn different tasks from multiple different domains and distributions, and it is the first experiment designed to show whether a fully connected network can perform multiple tasks.

In our third experiment, we train a convolutional network to perform both MNIST tasks from the previous experiment. This network is identical in architecture to all convolutional architectures trained previously, and once again this network is compared to two dedicated networks of the same architecture.

Our fourth experiment tests whether a single network can learn multiple tasks across an even wider array of input data types. For this experiment, we train a single fully-connected network to perform four tasks: both MNIST tasks, the Boston housing regression task, and an IMDB reviews sentiment classification task (Maas et al., 2011).

Our fifth experiment seeks to explore the feasibility of our methodologies in the context of more complicated tasks. For this experiment, we trained a single convolutional model to perform four tasks. The first three of these are the age, gender, and ethnicity prediction tasks from the UTK-Face dataset (Zhang et al., 2017). The fourth task is the CIFAR10 task (Krizhevsky et al.). This experimental network is tested against four dedicated networks of identical architectures.

Our final experiment tests our training procedure on a single-task transformer-based architecture, with a primary focus on testing the pruning and optimization capabilities of RSN2. For this experiment, we tested an unpruned transformer-based model's performance on the Reuters-21578, Distribution 1.0 dataset accessed via the UCI Machine Learning Repository (Dua & Graff, 2017).

## 3 RESULTS

In this section, we present the results for each experiment conducted. Firstly, we present the results for each control model for all tasks, as well as describe the model architectures and training schedules, as well as any data preprocessing steps which were taken. For control results, each control model was created only once and will therefore be presented with greater detail upon first presentation, but its results will be included in every pertinent experiment. For more detailed indicators of performance for all models, as well as mappings between integer labels and their respective classes, we refer the reader to the appendices.

### 3.1 EXPERIMENT 1: MNIST FASHION ENSEMBLE, CONVOLUTIONAL ARCHITECTURE

Our first experiment involved creating a single convolutional network with five disjoint subnetworks each trained on the MNIST Fashion dataset. We utilized a convolutional architecture with two convolutional blocks, with the first block containing two convolutional layers with thirty-two $3 \times 3$ filters and ReLU activation (Agarap, 2018) followed by maximum pooling over a $2 \times 2$ filter area. The second convolutional block was identical to the first, except each of the convolutional layers utilized sixty-four filters. The output to the convolutional blocks were then flattened and fed into two fully-connected layers with 256 neurons each and ReLU activation. A final fully-connected layer with ten neurons and softmax activation was used to provide the final outputs. The only preprocessing which was done on input data was a division of pixel values by 255 to ensure all input values were within the interval from 0 to 1. The model was trained using a batch size of 512, and early stopping was initialized after three epochs with no improvement of greater than 0.01 in loss on validation data. The same architecture, preprocessing steps, and training procedure were used in the experimental case as well, with each subnetwork pruned so that 20% of the network weights were allocated to each task. A summary of results can be found in Table 1.

Table 1: Experiment 1 Results.

| NETWORK | ACCURACY | PRUNING RATE |
|---|---|---|
| EXPERIMENTAL | | |
| SUBNETWORK 1 | 90% | 80% |
| SUBNETWORK 2 | 90% | 80% |
| SUBNETWORK 3 | 90% | 80% |
| SUBNETWORK 4 | 90% | 80% |
| SUBNETWORK 5 | 92% | 80% |
| ENSEMBLE | 92% | 80% EACH |
| CONTROL | | |
| MNIST FASHION | 92% | N/A |

Table 2: Experiment 2 results.

| NETWORK | ACCURACY | PRUNING RATE |
|---|---|---|
| EXPERIMENTAL | | |
| MNIST DIGIT | 97% | 90% |
| MNIST FASHION | 86% | 90% |
| CONTROL | | |
| MNIST DIGIT | 97% | N/A |
| MNIST FASHION | 88% | N/A |

## 3.2 EXPERIMENT 2: MNIST DIGITS AND MNIST FASHION MODEL, FULLY-CONNECTED ARCHITECTURE

Our second experiment is the first experiment to truly test whether a single model can learn disparate tasks across a variety of data modalities. To do this, we trained a single fully-connected network to perform both the MNIST Digit Recognition and the MNIST Fashion Recognition tasks. For this experiment, we preprocessed the data for both tasks once more by dividing each pixel value by $255$. Each image was then flattened into a two-dimensional vector. The images would then be passed through six fully connected layers each containing $1000$ artificial neurons each and activated using the ReLU activation function. The final layer contained ten neurons which were activated with the softmax activation function. The model was trained with the same early stopping conditions as all previous models, and a batch size of $512$ was used once more. The experimental model was trained using the same procedure and with the same architecture, with each task pruned to utilize 10% of the network weights. The results of this experiment can be found in Table 2.

## 3.3 EXPERIMENT 3: MNIST FASHION TWO-TASK MODEL, CONVOLUTIONAL ARCHITECTURE

For our third experiment, we utilized an identical architecture, data preprocessing, and training procedure to the previous experiment but trained two individual subnetworks within that architecture. We trained the first subnetwork on the five classes which were most easily identified by the control model, measured by F1 score, and we trained the second subnetwork on the five classes which were the most difficult for the control model to identify, also measured by F1 score. The results of the experiment can be found in Table 3.

## 3.4 EXPERIMENT 4: MNIST DIGIT, MNIST FASHION, BOSTON HOUSING, AND IMDB REVIEWS MODEL, FULLY-CONNECTED ARCHITECTURE

Our fourth experiment provides perhaps the most wide-ranging set of tasks for a single network, and as a result it is perhaps the best test of our multiple subnetwork hypothesis. In this experiment, we utilized a single fully-connected neural network and trained the network on four separate tasks; the first two of these tasks, both the MNIST Digit and MNIST Fashion tasks, were trained simultaneously. The third task, the Boston Housing regression task, was then trained. Finally, the IMDB reviews sentiment classification task was trained. The same architecture as previously used in all fully-connected experiments was used in this experiment. For the MNIST and Boston Housing

Table 3: Experiment 3 results.

| NETWORK | ACCURACY | PRUNING RATE |
|---|---|---|
| **EXPERIMENTAL** | | |
| EASY TASK | 98% | 80% |
| HARD TASK | 84% | 80% |
| **CONTROL** | | |
| MNIST FASHION | 92% | N/A |

Table 4: Experiment 4 results. In the performance column, percentages correspond to accuracy percentage on test data, while decimals represent mean squared error on test data.

| NETWORK | PERFORMANCE | PRUNING RATE |
|---|---|---|
| **EXPERIMENTAL** | | |
| MINST DIGIT | 97% | 90% |
| MNIST FASHION | 87% | 90% |
| BOSTON HOUSING | 0.011 | 90% |
| IMDB | 82% | 90% |
| **CONTROL** | | |
| MNIST DIGIT | 97% | N/A |
| MNIST FASHION | 88% | N/A |
| BOSTON HOUSING | 0.017 | N/A |
| IMDB | 80% | N/A |

tasks, the same preprocessing and reshaping efforts took place as previous experiments with fully-connected architectures. For the IMDB task, only the most common 10000 words were utilized and sequences were padded (or truncated) to 128 words, with both padding and truncating occurring at the end of the review. Furthermore, each of the input tokens was passed through an embedding layer, which embedded the token in a two-dimensional vector space. The embedded tensors were then flattened into a two-dimensional vector and passed through the architecture used throughout this experiment[1]. For the multitask model in this experiment, the first fully-connected layer which processed inputs from the IMDB task was dedicated solely to the IMDB task due to the differences in input shapes between all tasks. During training, the same early stopping criteria were used for all training iterations, batches sizes of 512 were used for the MNIST and IMDB tasks, while a batch size of 32 was used for the Boston Housing task. The results for this experiment can be found in Table 4, and task losses can be found in the appendices Section B. The model was pruned such that each task utilizes 10% of the available weights in fully-connected layers; the embedding layer was not pruned.

### 3.5 EXPERIMENT 5: UTKFACE AND CIFAR10 FOUR-TASK MODEL, CONVOLUTIONAL ARCHITECTURE

In our fifth experiment, we tested the RSN2 training procedure on a larger network with more complex tasks involved. We created a single network with a convolutional architecture on four individual tasks, three tasks from the UTKFace dataset (Zhang et al., 2017) and one multiclass classification task from CIFAR10 (Krizhevsky et al.). For the age task in the UTKFace data, we grouped values into classes by decade, excluding a final class containing all individuals with age greater than 90.

Architecturally, the model trained in this experiment contains three convolutional blocks consisting of 16, 32, and 64 $3 \times 3$ filters, respectively. Each block contains two convolutional layers with padding to preserve shape, followed by a max pooling layer. These blocks then feed into a fully connected architecture of three layers, each with 128 neurons each and activated using ReLU activation (Agarap, 2018), followed by an output layer of the required shape to perform the specified task.

For the multitask model, we utilized a shared convolutional embedding between all UTKFace tasks. This resulted in a two-task structure for convolutional layers within the model, with one task chan-

---

[1]This architecture involves six fully-connected layers of 1000 neurons and ReLU activation.

Table 5: Experiment 5 results.

| NETWORK | ACCURACY | PRUNING RATE |
|---|---|---|
| **EXPERIMENTAL** | | |
| UTKFACE AGE | 55% | 75% |
| UTKFACE GENDER | 89% | 75% |
| UTKFACE ETHNICITY | 77% | 75% |
| CIFAR10 | 50% | 75% |
| **CONTROL** | | |
| UTKFACE AGE | 32% | N/A |
| UTKFACE GENDER | 48% | N/A |
| UTKFACE ETHNICITY | 14% | N/A |
| CIFAR10 | 10% | N/A |

Table 6: Experiment 6 results.

| NETWORK | ACCURACY | PRUNING RATE |
|---|---|---|
| CONTROL | 75% | N/A |
| EXPERIMENTAL | 79% | 90% |

nel processing the UTKFace images and the other task channel processing the CIFAR10 images. For the fully-connected layers, each of the UTKFace tasks was isolated and a four-task architecture was used. Along both the convolutional and the fully-connected architectures, each task was pruned to utilize only $25\%$ of the model's available weights per task, meaning $50\%$ of the weights in the convolutional layers were utilized[2] and $100\%$ of the weights across the shared fully-connected architecture was utilized.[3] All four tasks were trained simultaneously, with early stopping criteria utilized with patience of five epochs. A summary of the results of this experiment can be found in Table 5. In addition to these performance results, which interestingly showed that the dedicated control models were unable to converge but that the multitask model was able to converge, it was also found that the multitask model only utilized $130MB$ of disk space when saved, while the combined single-task models utilized $203MB$ of disk space, over $56\%$ more than the multitask model.

### 3.6 EXPERIMENT 6: TRANSFORMER-BASED ARCHITECTURE

For our final experiment, we tested the applicability of our RSN2 training procedure in training a transformer-based architecture (Vaswani et al., 2017). For this experiment, we created a model with sequence input length of $512$ and a vocabulary of $30000$, 8 attention heads, a feed-forward dimension of $1028$, and a dropout rate of $10\%$. Both models have approximately $35M$ parameters. We trained our control and experimental models on the Reuters-21578, Distribution 1.0 dataset accessed via the UCI Machine Learning Repository (Dua & Graff, 2017) with a batch size of 256 and utilizing $20\%$ of the training dataset as hold-out validation data for early stopping.

For the experimental model, we utilized our multitask representational structure for feed-forward layers within the multi-headed attention components of the model, and these layers as well as all downstream fully-connected layers were pruned. Due to the complexity of the embedding layers within the model architectures, however, instead of applying pruning immediately before training, we allowed our experimental model to be trained for two epochs before applying pruning. Once this initial training occurred, we pruned the model using our gradient-based approach directly such that of $90\%$ of weights within the aforementioned pruned layers were active.

A summary of this experiment's results can be found in Table 6. In addition to these results, we also saw a significant reduction of the size of the resulting models. The experimental model required $133MB$ of disk space, while the control model required $397MB$, nearly $200\%$ more than the pruned model.

---

[2]This is due to the two-task structure of the convolutional layers, as mentioned above.
[3]This is due to the four-task structure of the fully-connected layers, as mentioned above.

### 3.7 LIMITATIONS

There are a number of limitations to this study which limit the conclusions we can draw from the results. Firstly, our experiments focus primarily on computer vision related tasks in this work, with all experiments containing some image classification task. While we do also include experiments which test our methods on both tabular input and natural language inputs, further studies should still be conducted to more rigorously test our hypotheses and methods on various input and data types.

Another limitation to this study is the limited size of both the datasets and models which were studied. Many state of the art models have been trained on millions of data samples, and the models themselves consist of many tens or hundreds of millions of parameters. Our study, on the other hand, considered labeled data with up to tens of thousands of samples and models which contained hundreds of thousands to low-tens-of-millions of parameters, orders of magnitude fewer (in both cases) than the state of the art in most cases. It is therefore unknown whether these methods would need to be modified[4] to accommodate larger modeling paradigms.

## 4 DISCUSSION AND CONCLUSIONS

This study was designed to test our multiple subnetwork hypothesis, which proposed that weights that might otherwise be pruned can be repurposed, ultimately yielding ANNs which can effectively perform multiple tasks across multiple data domains. To test our hypothesis, we developed the RSN2 training procedure to help identify subsets of weights within a network which are well-suited for a particular tasks. To combat CF, we then developed a customized ANN representational structure designed to isolate individual subnetworks within the overall architecture of the global model on a layer-by-layer basis. We then utilized these techniques in a series of experiments across a range of datasets utilizing a variety of common feedforward model architectures. , Across each of these experiments, we consistently saw that a single network was able to converge on multiple tasks by utilizing only a small fraction of network weights for each task. Furthermore, multitask models were able to perform each task with near identical performance to dedicated models with all weights active, even exceeding dedicated model performance in a few cases. Additionally, we saw that there needed to be no similarity between tasks, as our methodologies do not utilize transfer learning across tasks but instead rely on completely logically-isolated subnetworks within the network architecture. In our most extreme experiment, we were able to teach a single model to perform two separate image classification tasks and a sentiment classification task on natural language.

In addition to these results, we also saw performance improvements in terms of saved model sizes, which we recorded for our larger models in our final two experiments. While also maintaining performance compared to dedicated models for all tasks, our multitask convolutional model utilized only $64\%$ of the disk space of all of the dedicated, unpruned models. Similarly, our transformer-based model required only $33.5\%$ of the space the unpruned model required. All of these size reduction numbers were achieved without the explicit use of sparse tensors in TensorFlow (Abadi et al., 2015).

Despite the limitations to this study, there are still a number of conclusions we can draw from its results. Each multitask model was clearly able to converge on all its assigned tasks, achieving performance similar to, and in some cases exceeding, individual dedicated learners. These results therefore confirm our Multiple Subnetwork Hypothesis, showing that a single neural architecture is capable of utilizing small portions of itself to learn individual tasks. Additionally, we showed that our methodology for identifying and logically separating network weights is robust to CF, and we showed that we were therefore able to train multitask models across various domains and tasks both in parallel or in sequence without any adverse effects on previously-learned tasks.

Because the resulting model would have the ability to robustly perform multiple tasks, we can also conclude that a single model trained with these techniques could substitute multiple separate models in a deployed scenario. This substitution would therefore reduce resource requirements for organizations deploying and utilizing models, thus saving costs and simplifying the deployment architectures these organizations use. These models would also be able to acquire more downstream tasks over

---

[4]In particular, it is unknown whether our one-shot sparsification technique would generalize to much larger models.

their traditionally-trained counterparts after previously being trained on initial tasks, meaning adding new predictive analytics would be a much simpler undertaking as well, as it would only involve updating an already-deployed model and not require creating a completely new deployment for every new analytic.

## 4.1 FUTURE WORK

Following studies should be designed to address some of the limitations of this work as well as to improve the methods used within. Future studies should provide a more thorough analysis on how these methods perform on a larger variety of data modalities, including data types, task types, data availability, and model sizes. Secondly, we intend to identify whether a more precise pruning method can be applied as training occurs instead of in a one-shot manner as was done in this study. If pruning can be applied in a more intelligent manner, we believe the resulting model would potentially be able to more consistently surpass unpruned model performance while also achieving greater sparsification rates than possible with one-shot techniques.

Additionally, this study does not perform any analysis to identify the maximum amount of tasks which can be learned by a single network. Future work should therefore address finding the upper bounds of the number of tasks an individual model can support. Finally, the methods in this work do not allow for any transfer learning across tasks; instead, it is assumed that new tasks should utilize a completely disjoint subset of weights relative to all previously-learned tasks. Additional studies should be conducted to identify whether transfer learning can be leveraged during the acquisition of similar tasks to enable further reduction in the number of active weights and the quicker acquisition of new tasks.

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

## A    VISUAL REPRESENTATION OF RSN2

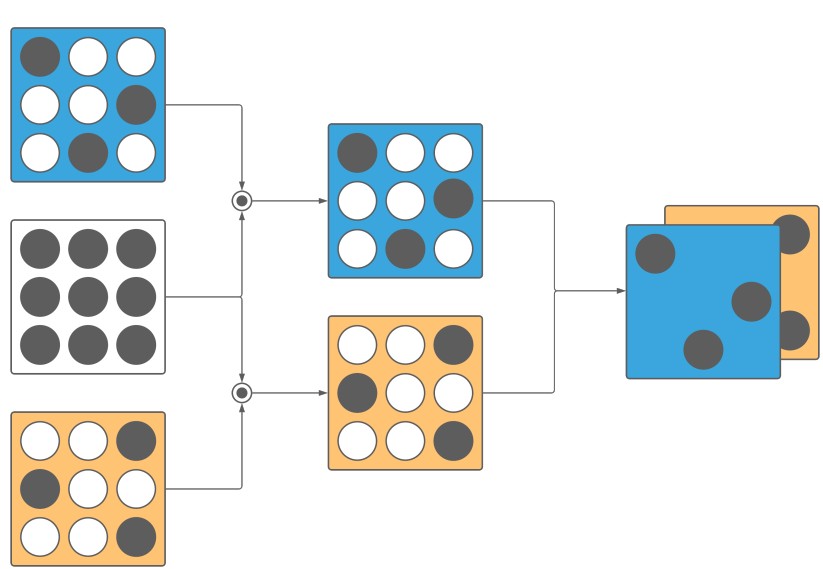

Figure 1: Diagram of the RSN2 training procedure and multitask representational structure for a single fully-connected layer with two tasks. On the left, we begin with the initialized layer weight matrix (center) and create two binary mask matrices (top and bottom). These binary masks are then bitwise-multiplied to the weight matrix, resulting in two disjoint subsets of weights which are then separated across the task dimension in our representational structure. Note to the right, no two corresponding parameters are nonzero.

# B TASK LOSSES, EXPERIMENT 4

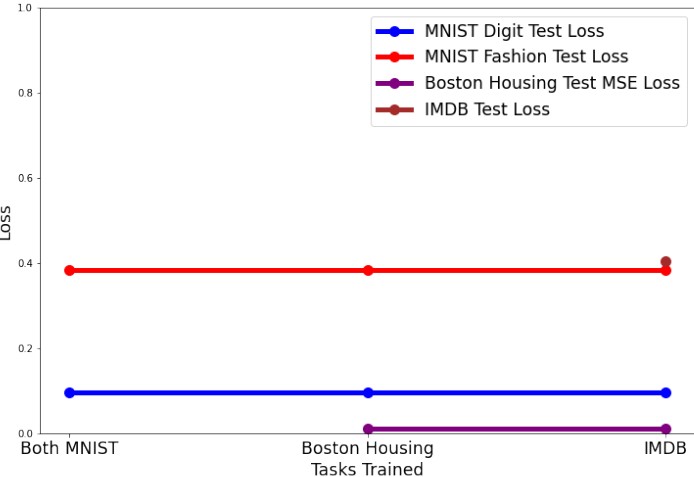

Figure 2: Task losses obtained on test data for multitask model after training each set of tasks.

## C CLASS LABELS FOR CLASSIFICATION TASKS.

Table 7: Class labels for MNIST tasks.

| DATASET | INTEGER | LABEL |
|---|---|---|
| MNIST DIGIT | | |
| | 1 | DIGIT "0" |
| | 2 | DIGIT "1" |
| | 3 | DIGIT "2" |
| | 4 | DIGIT "3" |
| | 5 | DIGIT "4" |
| | 6 | DIGIT "5" |
| | 7 | DIGIT "6" |
| | 8 | DIGIT "7" |
| | 9 | DIGIT "8" |
| | 10 | DIGIT "9" |
| MNIST FASHION | | |
| | 1 | T-SHIRT/TOP |
| | 2 | TROUSER |
| | 3 | PULLOVER |
| | 4 | DRESS |
| | 5 | COAT |
| | 6 | SANDAL |
| | 7 | SHIRT |
| | 8 | SNEAKER |
| | 9 | BAG |
| | 10 | ANKLE BOOT |

Table 8: Class labels for CIFAR10 and UTKFace tasks.

| Dataset | Integer | Label |
|---|---|---|
| CIFAR-10 | | |
| | 1 | Airplane |
| | 2 | Automobile |
| | 3 | Bird |
| | 4 | Cat |
| | 5 | Deer |
| | 6 | Dog |
| | 7 | Frog |
| | 8 | Horse |
| | 9 | Ship |
| | 10 | Truck |
| UTKFace Age | | |
| | 1 | $0 - 10$ |
| | 2 | $11 - 20$ |
| | 3 | $21 - 30$ |
| | 4 | $31 - 40$ |
| | 5 | $41 - 50$ |
| | 6 | $51 - 60$ |
| | 7 | $61 - 70$ |
| | 8 | $71 - 80$ |
| | 9 | $81 - 90$ |
| | 10 | $91+$ |
| UTKFace Gender | | |
| | 1 | Male |
| | 2 | Female |
| UTKFace Ethnicity | | |
| | 1 | White |
| | 2 | Black |
| | 3 | Asian |
| | 4 | Indian |
| | 5 | "Other" (Defined by dataset creators) |

Table 9: Class labels for IMDB task.

| Integer | Label |
|---|---|
| 1 | Negative Review |
| 2 | Positive Review |

# D    ADDITIONAL MODEL PERFORMANCE METRICS.

When conducting our study, we collected additional performance measures for all models which were not included in the main manuscript above. These metrics are presented in the following tables.

Table 10: Per-class F1 and overall accuracy scores for control computer vision models. "FC" indicates fully-connected architecture. "Conv" indicates convolutional architecture.

| NETWORK | CLASS NUMBER | | | | | | | | | | ACCURACY |
| | 1 | 2 | 3 | 4 | 5 | 6 | 7 | 8 | 9 | 10 | |
|---|---|---|---|---|---|---|---|---|---|---|---|
| MNIST DIGIT FC | .98 | .99 | .97 | .97 | .98 | .97 | .98 | .97 | .96 | .96 | .97 |
| MNIST DIGIT CONV | .99 | .99 | .98 | .99 | .99 | .99 | .99 | .99 | .99 | .98 | .99 |
| MNIST FASHION FC | .84 | .98 | .79 | .88 | .79 | .97 | .69 | .95 | .97 | .96 | .88 |
| MNIST FASHION CONV | .85 | .99 | .87 | .92 | .86 | .98 | .76 | .97 | .99 | .97 | .92 |
| CIFAR-10 CONV (EXP 6) | .55 | .59 | .35 | .26 | .38 | .39 | .51 | .53 | .55 | .52 | .47 |
| UTKFACE AGE CONV | 0 | 0 | .49 | 0 | 0 | 0 | 0 | 0 | 0 | 0 | .32 |
| UTKFACE GENDER CONV | 0 | 64 | N/A | N/A | N/A | N/A | N/A | N/A | N/A | N/A | 48 |
| UTKFACE ETHNICITY CONV | 0 | 0 | 25 | 0 | 0 | 18 | N/A | N/A | N/A | N/A | .14 |
| CIFAR-10 CONV (EXP 8) | 0 | 0 | 0 | 0 | 0 | 0 | 0 | 0 | 0 | 0 | 10 |

Table 11: Per-class F1 and overall accuracy scores for experimental computer vision models.

| Network | 1 | 2 | 3 | 4 | Class Number 5 | 6 | 7 | 8 | 9 | 10 | Accuracy |
|---|---|---|---|---|---|---|---|---|---|---|---|
| **Exp 1** | | | | | | | | | | | |
| Subnetwork 1 | 0.86 | 0.98 | 0.85 | 0.90 | 0.84 | 0.98 | 0.72 | 0.96 | 0.97 | 0.97 | 0.90 |
| Subnetwork 2 | 0.85 | 0.97 | 0.84 | 0.89 | 0.83 | 0.98 | 0.72 | 0.96 | 0.98 | 0.96 | 0.90 |
| Subnetwork 3 | 0.84 | 0.98 | 0.82 | 0.88 | 0.83 | 0.98 | 0.72 | 0.96 | 0.97 | 0.97 | 0.90 |
| Subnetwork 4 | 0.85 | 0.98 | 0.84 | 0.90 | 0.84 | 0.98 | 0.72 | 0.96 | 0.98 | 0.97 | 0.90 |
| Subnetwork 5 | 0.87 | 0.99 | 0.88 | 0.91 | 0.88 | 0.98 | 0.78 | 0.96 | 0.98 | 0.97 | 0.92 |
| Ensemble | 0.87 | 0.98 | 0.87 | 0.91 | 0.86 | 0.98 | 0.76 | 0.97 | 0.98 | 0.97 | 0.92 |
| **Exp 2** | | | | | | | | | | | |
| Easy Subnetwork | N/A | 1.00 | N/A | N/A | N/A | 0.97 | N/A | 0.96 | 0.99 | 0.96 | 0.98 |
| Hard Subnetwork | 0.85 | N/A | 0.87 | 0.92 | 0.85 | N/A | 0.73 | N/A | N/A | N/A | 0.84 |
| **Exp 3** | | | | | | | | | | | |
| MNIST Digit | 0.99 | 0.99 | 0.98 | 0.97 | 0.97 | 0.97 | 0.98 | 0.98 | 0.97 | 0.96 | 0.98 |
| MNIST Fashion | 0.82 | 0.97 | 0.77 | 0.87 | 0.79 | 0.96 | 0.62 | 0.94 | 0.97 | 0.95 | 0.87 |
| **Exp 4** | | | | | | | | | | | |
| MNIST Digit | 0.99 | 0.99 | 0.99 | 0.99 | 0.99 | 0.99 | 0.99 | 0.99 | 0.98 | 0.98 | 0.99 |
| MNIST Fashion | 0.85 | 0.98 | 0.85 | 0.90 | 0.85 | 0.98 | 0.71 | 0.96 | 0.97 | 0.96 | 0.90 |
| **Exp 5** | | | | | | | | | | | |
| MNIST Digit | 0.98 | 0.99 | 0.97 | 0.97 | 0.97 | 0.97 | 0.97 | 0.98 | 0.97 | 0.96 | 0.97 |
| MNIST Fashion | 0.82 | 0.98 | 0.78 | 0.88 | 0.78 | 0.95 | 0.68 | 0.94 | 0.96 | 0.95 | 0.87 |
| **Exp 6** | | | | | | | | | | | |
| MNIST Digit | 0.98 | 0.99 | 0.98 | 0.98 | 0.98 | 0.98 | 0.98 | 0.98 | 0.98 | 0.97 | 0.98 |
| MNIST Fashion | 0.84 | 0.98 | 0.83 | 0.89 | 0.83 | 0.97 | 0.70 | 0.95 | 0.97 | 0.96 | 0.89 |
| CIFAR-10 | 0.65 | 0.72 | 0.46 | 0.43 | 0.49 | 0.49 | 0.69 | 0.70 | 0.70 | 0.69 | 0.60 |
| **Exp 7** | | | | | | | | | | | |
| MNIST Digit | 0.98 | 0.99 | 0.98 | 0.96 | 0.97 | 0.96 | 0.97 | 0.96 | 0.96 | 0.96 | 0.97 |
| MNIST Fashion | 0.82 | 0.97 | 0.78 | 0.87 | 0.78 | 0.95 | 0.64 | 0.93 | 0.95 | 0.95 | 0.87 |
| **Exp 8** | | | | | | | | | | | |
| UTKF Age | 0.89 | 0.45 | 0.68 | 0.37 | 0.14 | 0.44 | 0.34 | 0.29 | 0.45 | 0.08 | 0.55 |
| UTKF Gender | 0.90 | 0.88 | N/A | N/A | N/A | N/A | N/A | N/A | N/A | N/A | 0.89 |
| UTKF Ethnicity | 0.84 | 0.83 | 0.79 | 0.68 | 0.27 | N/A | N/A | N/A | N/A | N/A | 0.77 |
| CIFAR-10 | 0.58 | 0.63 | 0.27 | 0.35 | 0.44 | 0.47 | 0.53 | 0.54 | 0.61 | 0.52 | 0.50 |

Table 12: Per-class F1 and overall accuracy scores for experimental and control IMDB sentiment analysis models.

| NETWORK | CLASS NUMBER | | ACCURACY |
|---|---|---|---|
| | 1 | 2 | |
| EXPERIMENT 7 IMDB | 0.83 | 0.81 | 0.82 |
| CONTROL IMDB | 0.80 | 0.80 | 0.80 |

Table 13: Boston Housing model losses.

| NETWORK | LOSS |
|---|---|
| EXPERIMENT 5 | 0.016 |
| EXPERIMENT 7 | 0.011 |
| CONTROL | 0.017 |

Table 14: Transformer performances. Per-class performance is reported as F1 score.

| CLASS NUMBER | CONTROL PERFORMANCE | EXPERIMENTAL PERFORMANCE |
|:---:|:---:|:---:|
| 1 | 0.48 | 0.70 |
| 2 | 0.80 | 0.75 |
| 3 | 0.48 | 0.63 |
| 4 | 0.92 | 0.93 |
| 5 | 0.83 | 0.85 |
| 6 | 0.00 | 0.00 |
| 7 | 0.57 | 0.96 |
| 8 | 0.00 | 0.00 |
| 9 | 0.41 | 0.69 |
| 10 | 0.80 | 0.88 |
| 11 | 0.88 | 0.90 |
| 12 | 0.64 | 0.73 |
| 13 | 0.00 | 0.40 |
| 14 | 0.59 | 0.56 |
| 15 | 0.00 | 0.40 |
| 16 | 0.14 | 0.17 |
| 17 | 0.70 | 0.74 |
| 18 | 0.33 | 0.32 |
| 19 | 0.45 | 0.56 |
| 20 | 0.66 | 0.68 |
| 21 | 0.41 | 0.55 |
| 22 | 0.54 | 0.67 |
| 23 | 0.00 | 0.00 |
| 24 | 0.12 | 0.24 |
| 25 | 0.50 | 0.49 |
| 26 | 0.54 | 0.76 |
| 27 | 0.75 | 0.35 |
| 28 | 0.00 | 0.67 |
| 29 | 0.20 | 0.37 |
| 30 | 0.00 | 0.33 |
| 31 | 0.33 | 0.32 |
| 32 | 0.32 | 0.50 |
| 33 | 0.24 | 0.95 |
| 34 | 0.00 | 0.80 |
| 35 | 0.35 | 0.73 |
| 36 | 0.00 | 0.50 |
| 37 | 0.14 | 0.70 |
| 38 | 0.00 | 0.00 |
| 39 | 0.50 | 0.00 |
| 40 | 0.00 | 0.13 |
| 41 | 0.00 | 0.46 |
| 42 | 0.46 | 0.62 |
| 43 | 0.00 | 0.00 |
| 44 | 0.00 | 0.00 |
| 45 | 0.67 | 0.89 |
| 46 | 0.00 | 0.67 |
| OVERALL ACCURACY | 0.75 | 0.79 |

