# OpenReview forum: "The Multiple Subnetwork Hypothesis: Enabling Multidomain Learning by Isolating Task-Specific Subnetworks in Feedforward Neural Networks"
_ICLR.cc/2023/Conference — Submitted to ICLR 2023_

### Official Review · Reviewer_aGRW · 2022-10-20

**Confidence:** 4
**Correctness:** 2
**Technical Novelty And Significance:** 3
**Empirical Novelty And Significance:** 2
**Recommendation:** 3

**Clarity, Quality, Novelty And Reproducibility:**

The idea behind the work - the concept that a neural network requires only a fraction of its weights to perform well - is one that has already been explored at length in the literature. The observation that this could allow a single model to be trained multiple times using disjoint subnetworks is wholly novel, however (at least as far as I am aware).
While I think the idea is interesting, the actual value of the contribution is not clear to me (see above for my specific concerns). I also have a number of concerns about their methodology - specifically the lack of any kind of comparison to other similar approaches, as well as the fact that each experiment was only a single trial rather than an average of multiple trials (or ideally an actual statistical significance test). Some of the results were also not reported clearly (see the previous discussion about table 3), which made them entirely inconclusive.
The clarity of the work was also highly lacking in parts, with the exact experimental settings for some of the experiments (notably experiment 6) being highly unclear. It would also have been helpful if the authors more clearly stated what their proposed contribution was (is it just model compression?), as well as a clearer case for why this model would actually lead to improved performance, if that is being claimed as a benefit of the approach.

**Strength And Weaknesses:**

This paper makes an interesting observation - if a network requires only a fraction of its weights to perform well, can the others be put to use for other tasks? The idea of performing multi-task learning this way is certainly interesting, and in a few cases they even show slightly improved performance with it over the control. I like the idea behind the approach, and I think a thorough evaluation of using ensembles of models trained in this way to compare performance against the base model could lead in interesting directions. I do have a number of reservations, however.

While this method is interesting, I am not sure what concrete benefit it actually offers. The ability to store multiple trained networks within one network without using additional memory is certainly interesting, but all of those networks are constrained to using the same architecture, and since there is no interaction between the separate tasks it is hard to see how there could be any type of transfer learning or improved performance. The memory savings of this approach are certainly interesting, but it is unclear to me whether how this memory/energy savings compares to the memory/energy savings of existing methods for pruning neural networks. Perhaps the benefit is that existing weight pruning methods require dedicated hardware/software for dealing with sparse matrices to be fully realized? If so however, this is not made fully clear in the paper, nor is it clear that the proposed architecture would avoid having the same limitations.
As such, I do not fully understand what the actual contribution here is meant to be. Since there is no comparison done against any other similar approaches for memory/energy savings, it is hard to see if this is a significant contribution. And since each subnetwork is entirely disjoint, I do not see how this method would be likely to lead to improved performance. The results shown are largely comparable to the control in accuracy, save for a few examples where the proposed approach leads to slight improvements - but there is no statistical analysis done to show whether these small improvements are significant or not. There is also no comparison done to the performance of other weight pruning approaches on the same tasks and architectures, either in terms of memory use or model accuracy.

The most compelling example given is experiment 6, using a transformer architecture. The setting of this experiment is highly unclear to me, however. What is the actual task being performed here - simply text classification? If so, how is the multi-task approach being used, as there is only one task under consideration? Is the multi-task model using an ensemble of models each learning to perform the same task? Are they split by class somehow? How many models are being used? Is the final result shown in table 6 an ensemble across multiple models, or an average, or something else? The table and description make it appear as though only one model is being used, but the proposed approach is explicitly a method for training a single model's weights on multiple tasks.

As an additional note, the table given for experiment 3 seems highly incomplete, as there is no way to directly compare the authors' model to the baseline. The table should include either the results of the baseline model split across the two task settings, or some sort of ensemble of the experimental model evaluated on the whole dataset. As is, the values in the table are meaningless as there is no way to compare them.



**Summary Of The Paper:**

This paper builds on existing work showing that artificial neural networks often require only a small fraction of their parameters to perform well. The paper proposes the 'Multiple Subnetwork Hypothesis', which extends this observation to conclude that the weights of a single network could be separated into disjoint subnetworks that could each be trained for separate tasks. This allows a single network to be trained on many different tasks without using increased memory or suffering from catastrohpic forgetting. The authors propose an algorithm called RSN2 to train a single network in this disjoint fashion. This algorithm works by copying the model weights across an additional 'task' dimension, then ensuring that only one weight can be active at a time across the task dimension by masking the weights based on an initial pruning performed at the beginning of training. They test their hypothesis in a number of multitask settings by comparing the performance of a control model on a single task to their multiple subnetwork model across many tasks, and show comparable performance to the control with significant memory savings.

**Summary Of The Review:**

Overall I think this paper proposed an interesting idea, but failed to make a convincing case for why it would actually be useful. There were no empirical comparisons done to other existing approaches, the results were not statistically significant, and the authors were not clear about what the intended benefit of their approach was and how it compared to other similar methods. Many of the experiment settings and tables were unclear, and I think it is difficult to conclude much about their approach from the results they presented. I think the paper would need significant rewriting including at a minimum:
a) direct comparisons of the memory/energy savings of their method to other existing approaches
b) justification for why they expect a performance increase from their method (if this is a claimed benefit)
c) experiments with multiple trials, including comparisons to other weight pruning approaches and proper comparisons against the baseline
d) a clearer description of the experimental settings

---

### Official Review · Reviewer_PPCR · 2022-10-20

**Confidence:** 4
**Clarity, Quality, Novelty And Reproducibility:** This paper provides sufficient implem…
**Correctness:** 3
**Technical Novelty And Significance:** 3
**Empirical Novelty And Significance:** 2
**Recommendation:** 5

**Strength And Weaknesses:**

Strengths:
1. The multiple subnetwork hypothesis is very interesting. The proposed multi-task training method based on this hypothesis is reasonable.
2. This paper provides sufficient and detailed implementation details. The pseudocode in Algorithm 1 is clear.
3. The limitations of the proposed method and future work are clearly discussed.

Weaknesses:
1. This paper evaluates the proposed method on both convolutional networks and transformer architectures. However, all the considered architectures do not belong to any popular architecture family. It would be better to conduct experiments on some popular architectures, e.g., ResNet [A] and Swin Transformer [B].

2. It is unclear what the essential differences are between the proposed RSN2 and existing pruning methods. Moreover, the comparisons among these methods are still missing.

3. This paper mainly conducts experiments on small datasets, such as MNIST and CIFAR. It would be better to consider some large-scale datasets, e.g., ImageNet.

4. As mentioned in the paper, some weights of subnetworks may be unnecessary for one task but can be used to learn other tasks. In other words, the performance of subnetworks may vary a lot among different tasks. However, such differences are still unclear in the paper. It would be interesting to visualize the performance distribution of subnetworks on different tasks, similar to what is shown in [C]. Moreover, it is also very interesting to see how large the ranking of subnetworks changes from one task to another one (e.g., the best subnetwork on one task may be only located at the median on another task).


5. As mentioned in Section 1, the main goal is to“negate the high computational cost of parameter isolation”. Nevertheless, it seems that there are no results to show how much computational cost can be reduced.

[A] Deep Residual Learning for Image Recognition, CVPR 2016.

[B] Swin Transformer: Hierarchical Vision Transformer Using Shifted Windows, ICCV 2021.

[C] Improving Robustness by Enhancing Weak Subnets, ECCV 2022.



**Summary Of The Paper:**

The authors proposed a hypothesis and network structure that allows a pruned network to employ previously unused weights to learn subsequent tasks. Specifically, the authors develop a customized ANN representational structure to isolate individual sub-networks and devise a training procedure named Reduction of Sub-Network Neuroplasticity (RSN2) to train the proposed model. Experimental results show that the proposed method can train an ANN without sacrificing performance or catastrophic forgetting.


**Summary Of The Review:**

The idea is interesting. The experiments can be further improved.

---

### Official Review · Reviewer_3BZv · 2022-10-24

**Confidence:** 3
**Correctness:** 3
**Technical Novelty And Significance:** 1
**Empirical Novelty And Significance:** 1
**Recommendation:** 3

**Clarity, Quality, Novelty And Reproducibility:**

The paper is poorly written with limited novelty:
- There are not sufficient related work comparison including the lottery ticket, and sparsely activated mixture-of-expert networks. The key points are very similar.
- The evaluation is not clearly explained and only contains small datasets.

**Strength And Weaknesses:**

The proposed idea looks very similar to lottery ticket hypothesis [1], such that there is a winning ticket for every task. The problem formulation, the expanded kernel tensor, looks very much like a mixture-of-expert layer while the sparsification and subnetwork identification are very much like the routing function used in a mixture-of-expert network. In this sense, this paper has very limited novelty and add-on contribution compared to related work. At least, the paper should explain why it is different from related work and why this method address multitask learning better.

Evaluations are not through by only comparing on tiny datasets like MNIST. The tables are extremely hard to interpret.

[1]: https://arxiv.org/abs/1803.03635

**Summary Of The Paper:**

The paper proposed a method that enables a pruned network to employ previously unused weights to learn subsequent tasks. The exploration along this work leads to multiple subnetwork hypothesis-- that a dense, randomly initialized feedforward ANN contains within its architecture multiple disjoint subnetworks which can be utilized together to learn and make accurate predictions on multiple tasks.

**Summary Of The Review:**

The paper needs significant improvement before being accepted by any major conferences. First, please elaborate why this method is different from traditional approaches like lifelong learning with lottery ticket [1], and multitask learning with sparsely gated mixture-of-experts. In a MoE network, one or a few experts are selected for each data, which is a more general formulation that subsumes the proposed method in this paper. The paper should consider improving the baselines and evaluation benchmarks. For example, how does this proposed method improve against traditional lifelong learning methods and multitask methods. If the emphasis is on lifelong learning, then the paper should focus on a streaming of tasks.  If multitask is the priority, then should compare against approaches with stronger performance on multitask learning, and sparsely gated MoE can be a good one.

[1]: https://openreview.net/forum?id=LXMSvPmsm0g

---

### Official Review · Reviewer_yCzy · 2022-10-27

**Confidence:** 4
**Correctness:** 2
**Technical Novelty And Significance:** 1
**Empirical Novelty And Significance:** 2
**Recommendation:** 3

**Clarity, Quality, Novelty And Reproducibility:**

## Clarity
The clarity of the paper is poor in particular in the Methodology section.  In addition, for the experiments, it's unclear what the control set of results corresponds to.
## Quality
The quality of the experiments is low since there is no discussion of how many seeds were run or any notion of variance of the results.  Additionally, there was no comparison to meaningful competitors beyond the control. One specific question I have is why the result for CIFAR-10 in Table 5 is no better than random at 10%.
## Reproducibility
The authors do not provide enough detail about the experiments to reproduce results.  Experiments also appear to be run just once without any measure of variance/significance.

**Strength And Weaknesses:**

Pros:
- Perhaps unsurprisingly, the authors are able to demonstrate that disjoint subsets of network weights can be used for different tasks without loss in performance.
- The approach to more efficient multi-task learning with a single network is simple.

Cons:
- The authors do not adequately discuss related work in multi-task learning.  Learning Sparse Sharing Architectures for Multiple Tasks by Sun et al., 2019 is especially relevant and not discussed at all.
- The proposed approach is not very interesting since there is no knowledge sharing across tasks by design with disjoint subset of weights to avoid catastrophic forgetting. There is obvious benefit to sharing as recognized by the authors in Experiment 5 in which the UTKFace tasks share convolutional embeddings.

**Summary Of The Paper:**

This paper builds upon the lottery ticket hypothesis and posits the multiple subnetwork hypothesis in which weights of a single network are disjointly allocated to multiple tasks to avoid catastrophic forgetting while achieving competitive task-wise performance on all tasks.

**Summary Of The Review:**

I recommend reject for this paper due to the lack of novelty and insight in the proposed method for using disjoint weights of a single network to learn multiple tasks. There is no sharing at all between tasks unless introduced in an ad-hoc manner manually for specific tasks.  The experiments are also limited and of low quality.

---

### Decision · Program_Chairs · 2023-01-20

**Decision:**

Reject

**Justification For Why Not Higher Score:**

The unanimity of the rejection is due to the unclarity of the presentation as well as the incompleteness of the empirical analysis. Yet, the idea is novel, interesting, and could be useful to the community of representation learning. If suggestions provided by the reviewers are taken into account for future submissions, this would be a very strong case for acceptance and valuable work to the whole community.

**Justification For Why Not Lower Score:**

N/A

**Metareview: Summary, Strengths And Weaknesses:**

I Summary, Strengths, and Weaknesses

I Summary:

I.1 Investigated Problem:

The paper builds upon the lottery ticket hypothesis and presents the multiple subnetwork hypothesis. The hypothesis considers the observation that artificial neural networks often require only a small fraction of their parameters to perform well to arrive at a conclusion stating that the weights of a single network could be separated into disjoint subnetworks that could each be trained for separate tasks.

- I.2 Proposed Solution:

This allows a single network to be trained on many different tasks without using increased memory or suffering from catastrophic forgetting.

- I.3 Validity Proof of the Proposed Solution:

The hypothesis is tested for a number of multi-task settings by comparing the performance of a control model on a single task to their multiple subnetwork model across many tasks, and results show comparable performance to the control with significant memory savings.

II Strengths:

- II.1 From a structural (organization) point of view:
    - Refer to III as from a structural point of view, there is a quasi-unanimity on the poor quality of the writing.

- II.2 From an analytical (development) point of view:
    - Using a single model to be trained multiple times using disjoint subnetworks is wholly novel idea that could be of great benefit to the whole community of representation learning;
    - The proposed method to perform multi-task learning is interesting and intuitive;
    - Sufficient details are provided when it comes to the implementation details;
    - Limitations of the proposed method and future work are clearly discussed which adds value to the submission.

- II.3 From a perspective of soundness (unity, and coherence) and completeness (correctness):
    - Here is where the authors failed to make a convincing case for their submission as there are shortcomings in terms of clarity and presentation of the method as well as the incompleteness of the empirical analysis.

III Addressing what can be thought of as weaknesses:

- III.1 From a structural point of view:
    - Unfortunately, there is a quasi-unanimity among reviewers about the poor quality of the written text and its clarity. It might be that the authors rushed their submission and the incompleteness of experimental analysis reinforces this presumption. In any case, the submission allows the authors to get valuable feedback from the reviewers so that they can build upon it to make a convincing case for their proposed solution. Inherently, the exposed idea is very interesting and worth investigating as most of the reviewers emphasized. The multiple subnetwork hypothesis can be of benefit to the community.

III.2 From an analytical (Development) point of view:
- No adequate discussion about related work in multi-task learning:
    - Related work with key similarities, such as the lottery ticket and sparsely activated mixture-of-expert are not compared to the proposed method. => Comparisons of the memory/energy savings of the method to other existing approaches, when it is possible, could be done for a future submission.
    - Results were not statistically significant:
        - each experiment was only a single trial rather than an average of multiple trials (or ideally an actual statistical significance test).
    - Contribution is not clearly stated:
        - It can be helpful if the authors clearly state what their proposed contribution is and investigate the reason behind the success of the presented solution.
    - Experiment settings and tables were unclear.

The feedback from the reviewers is really appreciated as they identified the strengths and weaknesses of the proposal and came up with valuable suggestions that can help the authors build a convincing case for their future submissions:

IV. Potential of the paper:

IV.1 From a Potential perspective:
It is clear that the idea presented has concrete potential because it could be very beneficial for the whole community of representation learning. The authors simply failed to convincingly prove the usefulness of the method. The incomplete analysis of the experiments and the lack of clarity in the submission did not help the reviewers appreciate the quality of their work. If rectified, the future submission may be a strong case for acceptance.